# Temporal Regularization in Markov Decision Process

**Pierre Thodoroff**
McGill University
pierre.thodoroff@mail.mcgill.ca

**Audrey Durand**
McGill University
audrey.durand@mcgill.ca

**Joelle Pineau**
McGill University & Facebook AI Research
jpineau@cs.mcgill.ca

**Doina Precup**
McGill University
dprecup@cs.mcgill.ca

## Abstract

Several applications of Reinforcement Learning suffer from instability due to high variance. This is especially prevalent in high dimensional domains. Regularization is a commonly used technique in machine learning to reduce variance, at the cost of introducing some bias. Most existing regularization techniques focus on spatial (perceptual) regularization. Yet in reinforcement learning, due to the nature of the Bellman equation, there is an opportunity to also exploit temporal regularization based on smoothness in value estimates over trajectories. This paper explores a class of methods for temporal regularization. We formally characterize the bias induced by this technique using Markov chain concepts. We illustrate the various characteristics of temporal regularization via a sequence of simple discrete and continuous MDPs, and show that the technique provides improvement even in high-dimensional Atari games.

## 1 Introduction

There has been much progress in Reinforcement Learning (RL) techniques, with some impressive success with games [30], and several interesting applications on the horizon [17, 29, 26, 9]. However RL methods are too often hampered by high variance, whether due to randomness in data collection, effects of initial conditions, complexity of learner function class, hyper-parameter configuration, or sparsity of the reward signal [15]. Regularization is a commonly used technique in machine learning to reduce variance, at the cost of introducing some (smaller) bias. Regularization typically takes the form of smoothing over the observation space to reduce the complexity of the learner's hypothesis class.

In the RL setting, we have an interesting opportunity to consider an alternative form of regularization, namely *temporal regularization*. Effectively, temporal regularization considers smoothing over the trajectory, whereby the estimate of the value function at one state is assumed to be related to the value function at the state(s) that typically occur before it in the trajectory. This structure arises naturally out of the fact that the value at each state is estimated using the Bellman equation. The standard Bellman equation clearly defines the dependency between value estimates. In temporal regularization, we amplify this dependency by making each state depend more strongly on estimates of *previous* states as opposed to multi-step methods that considers future states.

This paper proposes a class of temporally regularized value function estimates. We discuss properties of these estimates, based on notions from Markov chains, under the policy evaluation setting, and extend the notion to the control case. Our experiments show that temporal regularization effectively reduces variance and estimation error in discrete and continuous MDPs. The experiments also

highlight that regularizing in the time domain rather than in the spatial domain allows more robustness to cases where state features are mispecified or noisy, as is the case in some Atari games.

## 2   Related work

Regularization in RL has been considered via several different perspectives. One line of investigation focuses on regularizing the features learned on the state space [11, 25, 24, 10, 21, 14]. In particular backward bootstrapping method's can be seen as regularizing in feature space based on temporal proximity [34, 20, 1]. These approaches assume that nearby states in the state space have similar value. Other works focus on regularizing the changes in policy directly. Those approaches are often based on entropy methods [23, 28, 2]. Explicit regularization in the temporal space has received much less attention. Temporal regularization in some sense may be seen as a "backward" multi-step method [32]. The closest work to ours is possibly [36], where they define natural value approximator by projecting the previous states estimates by adjusting for the reward and $\gamma$. Their formulation, while sharing similarity in motivation, leads to different theory and algorithm. Convergence properties and bias induced by this class of methods were also not analyzed in Xu et al. [36].

## 3   Technical Background

### 3.1   Markov chains

We begin by introducing discrete Markov chains concepts that will be used to study the properties of temporally regularized MDPs. A discrete-time Markov chain [19] is defined by a discrete set of states $\mathcal{S}$ and a transition function $\mathcal{P} : \mathcal{S} \times \mathcal{S} \mapsto [0, 1]$ which can also be written in matrix form as $P_{ij} = \mathcal{P}(i|j)$. Throughout the paper, we make the following mild assumption on the Markov chain:

**Assumption 1.** *The Markov chain P is ergodic: P has a unique stationary distribution $\mu$.*

In Markov chains theory, one of the main challenge is to study the mixing time of the chain [19]. Several results have been obtained when the chain is called reversible, that is when it satisfies detailed balance.

**Definition 1** (Detailed balance [16]). *Let P be an irreducible Markov chain with invariant stationary distribution $\mu$[1]. A chain is said to satisfy detailed balance if and only if*

$$\mu_i P_{ij} = \mu_j P_{ji} \qquad \forall i, j \in \mathcal{S}. \tag{1}$$

Intuitively this means that if we start the chain in a stationary distribution, the amount of probability that flows from $i$ to $j$ is equal to the one from $j$ to $i$. In other words, the system must be at equilibrium. An intuitive example of a physical system not satisfying detailed balance is a snow flake in a coffee. Indeed, many chains do not satisfy this detailed balance property. In this case it is possible to use a different, but related, chain called the reversal Markov chain to infer mixing time bounds [7].

**Definition 2** (Reversal Markov chain [16]). *Let $\widetilde{P}$ the reversal Markov chain of P be defined as:*

$$\widetilde{P_{ij}} = \frac{\mu_j P_{ji}}{\mu_i} \qquad \forall i, j \in \mathcal{S}. \tag{2}$$

*If P is irreducible with invariant distribution $\mu$, then $\widetilde{P}$ is also irreducible with invariant distribution $\mu$.*

The reversal Markov chain $\widetilde{P}$ can be interpreted as the Markov chain $P$ with time running backwards. If the chain is reversible, then $P = \widetilde{P}$.

### 3.2   Markov Decision Process

A Markov Decision Process (MDP), as defined in [27], consists of a discrete set of states $\mathcal{S}$, a transition function $\mathcal{P} : \mathcal{S} \times \mathcal{A} \times \mathcal{S} \mapsto [0, 1]$, and a reward function $r : \mathcal{S} \times \mathcal{A} \mapsto \mathbb{R}$. On each round $t$, the learner observes current state $s_t \in \mathcal{S}$ and selects action $a_t \in \mathcal{A}$, after which it receives reward $r_t = r(s_t, a_t)$ and moves to new state $s_{t+1} \sim \mathcal{P}(\cdot|s_t, a_t)$. We define a stationary policy $\pi$ as a probability distribution over actions conditioned on states $\pi : \mathcal{S} \times \mathcal{A} \mapsto [0, 1]$.

### 3.2.1 Discounted Markov Decision Process

When performing policy evaluation in the discounted case, the goal is to estimate the discounted expected return of policy $\pi$ at a state $s \in \mathcal{S}$, $v^\pi(s) = \mathbb{E}_\pi[\sum_{t=0}^\infty \gamma^t r_{t+1}|s_0 = s]$, with discount factor $\gamma \in [0,1)$. This $v^\pi$ can be obtained as the fixed point of the Bellman operator $\mathcal{T}^\pi$ such that:

$$\mathcal{T}^\pi v^\pi = r^\pi + \gamma P^\pi v^\pi, \tag{3}$$

where $P^\pi$ denotes the $|\mathcal{S}| \times |\mathcal{S}|$ transition matrix under policy $\pi$, $v^\pi$ is the state values column-vector, and $r$ is the reward column-vector. The matrix $P^\pi$ also defines a Markov chain.

In the control case, the goal is to find the optimal policy $\pi^*$ that maximizes the discounted expected return. Under the optimal policy, the optimal value function $v^*$ is the fixed point of the non-linear optimal Bellman operator:

$$\mathcal{T}^* v^* = \max_{a \in \mathcal{A}}[r(a) + \gamma P(a)v^*]. \tag{4}$$

## 4  Temporal regularization

Regularization in the feature/state space, or *spatial regularization* as we call it, exploits the regularities that exist in the observation (or state). In contrast, *temporal regularization* considers the temporal structure of the value estimates through a trajectory. Practically this is done by smoothing the value estimate of a state using estimates of states that occurred earlier in the trajectory. In this section we first introduce the concept of temporal regularization and discuss its properties in the policy evaluation setting. We then show how this concept can be extended to exploit information from the entire trajectory by casting temporal regularization as a time series prediction problem.

Let us focus on the simplest case where the value estimate at the current state is regularized using only the value estimate at the previous state in the trajectory, yielding updates of the form:

$$v_\beta(s_t) = \mathbb{E}_{s_{t+1}, s_{t-1} \sim \pi}[r(s_t) + \gamma((1-\beta)v_\beta(s_{t+1}) + \beta v_\beta(s_{t-1}))]$$

$$= r(s_t) + \gamma(1-\beta) \sum_{s_{t+1} \in \mathcal{S}} p(s_{t+1}|s_t)v_\beta(s_{t+1}) + \gamma\beta \sum_{s_{t-1} \in \mathcal{S}} \frac{p(s_t|s_{t-1})p(s_{t-1})}{p(s_t)} v_\beta(s_{t-1}), \tag{5}$$

for a parameter $\beta \in [0,1]$ and $p(s_{t+1}|s_t)$ the transition probability induced by the policy $\pi$. It can be rewritten in matrix form as $v_\beta = r + \gamma(((1-\beta)P^\pi + \beta\widetilde{P^\pi})v_\beta)$, where $\widetilde{P^\pi}$ corresponds to the reversal Markov chain of the MDP. We define a temporally regularized Bellman operator as:

$$\mathcal{T}_\beta^\pi v_\beta = r + \gamma((1-\beta)P^\pi v_\beta + \beta\widetilde{P^\pi}v_\beta). \tag{6}$$

To alleviate the notation, we denote $P^\pi$ as $P$ and $\widetilde{P^\pi}$ as $\widetilde{P}$.

**Remark.** *For $\beta = 0$, Eq. 6 corresponds to the original Bellman operator.*

We can prove that this operator has the following property.

**Theorem 1.** *The operator $\mathcal{T}_\beta^\pi$ has a unique fixed point $v_\beta^\pi$ and $\mathcal{T}_\beta^\pi$ is a contraction mapping.*

*Proof.* We first prove that $\mathcal{T}_\beta^\pi$ is a contraction mapping in $L_\infty$ norm. We have that

$$\left\|\mathcal{T}_\beta^\pi u - \mathcal{T}_\beta^\pi v\right\|_\infty = \left\|r + \gamma((1-\beta)Pu + \beta\widetilde{P}u) - (r + \gamma((1-\beta)Pv + \beta\widetilde{P}v))\right\|_\infty$$

$$= \gamma\left\|((1-\beta)P + \beta\widetilde{P})(u-v)\right\|_\infty \tag{7}$$

$$\leq \gamma\|u-v\|_\infty,$$

where the last inequality uses the fact that the convex combination of two row stochastic matrices is also row stochastic (the proof can be found in the appendix). Then using Banach fixed point theorem, we obtain that $v_\beta^\pi$ is a unique fixed point. □

Furthermore the new induced Markov chain $(1-\beta)P + \beta\widetilde{P}$ has the same stationary distribution as the original $P$ (the proof can be found in the appendix).

**Lemma 1.** *$P$ and $(1-\beta)P + \beta\widetilde{P}$ have the same stationary distribution $\mu$   $\forall \beta \in [0,1]$.*

In the policy evaluation setting, the bias between the original value function $v^\pi$ and the regularized one $v_\beta^\pi$ can be characterized as a function of the difference between $P$ and its Markov reversal $\widetilde{P}$, weighted by $\beta$ and the reward distribution.

**Proposition 1.** *Let $v^\pi = \sum_{i=0}^\infty \gamma^i P^i r$ and $v_\beta^\pi = \sum_{i=0}^\infty \gamma^i ((1-\beta)P + \beta\widetilde{P})^i r$. We have that*

$$\left\| v^\pi - v_\beta^\pi \right\|_\infty = \left\| \sum_{i=0}^\infty \gamma^i (P^i - ((1-\beta)P + \beta\widetilde{P})^i)r \right\|_\infty \leq \sum_{i=0}^\infty \gamma^i \left\| (P^i - ((1-\beta)P + \beta\widetilde{P})^i)r \right\|_\infty . \tag{8}$$

*This quantity is naturally bounded for $\gamma < 1$.*

**Remark.** *Let $P^\infty$ denote a matrix where columns consist of the stationary distribution $\mu$. By the property of reversal Markov chains and lemma 1, we have that $\lim_{i \to \infty} \| P^i r - P^\infty r \| \to 0$ and $\lim_{i \to \infty} \| ((1-\beta)P + \beta\widetilde{P})^i r - P^\infty r \| \to 0$, such that the Marvov chain $P$ and its reversal $(1-\beta)P + \beta\widetilde{P}$ converge to the same value. Therefore, the norm $\| (P^i - ((1-\beta)P + \beta\widetilde{P})^i)r \|_p$ also converges to 0 in the limit.*

**Remark.** *It can be interesting to note that if the chain is reversible, meaning that $P = \widetilde{P}$, then the fixed point of both operators is the same, that is $v^\pi = v_\beta^\pi$.*

**Discounted average reward case:**   The temporally regularized MDP has the same discounted average reward as the original one as it is possible to define the discounted average reward [35] as a function of the stationary distribution $\pi$, the reward vector and $\gamma$ . This leads to the following property (the proof can be found in the appendix).

**Proposition 2.** *For a reward vector r, the MDPs defined by the the transition matrices $P$ and $(1-\beta)P + \beta\widetilde{P}$ have the same average reward $\rho$.*

Intuitively, this means that temporal regularization only reweighs the reward on each state based on the Markov reversal, while preserving the average reward.

**Temporal Regularization as a time series prediction problem:**   It is possible to cast this problem of temporal regularization as a time series prediction problem, and use richer models of temporal dependencies, such as exponential smoothing [12], ARMA model [5], etc. We can write the update in a general form using $n$ different regularizers $(\widetilde{v_0}, \widetilde{v_1}...\widetilde{v_{n-1}})$:

$$v(s_t) = r(s) + \gamma \sum_{i=0}^{n-1} [\beta(i)\widetilde{v}_i(s_{t+1})], \tag{9}$$

where $\widetilde{v}_0(s_{t+1}) = v(s_{t+1})$ and $\sum_{i=0}^{n-1} \beta(i) = 1$. For example, using exponential smoothing where $\widetilde{v}(s_{t+1}) = (1-\lambda)v(s_{t-1}) + (1-\lambda)\lambda v(s_{t-2})...$, the update can be written in operator form as:

$$\mathcal{T}_\beta^\pi v = r + \gamma \left( (1-\beta)Pv + \beta(1-\lambda) \sum_{i=1}^\infty \lambda^{i-1} \widetilde{P}^i v \right), \tag{10}$$

and a similar argument as Theorem 1 can be used to show the contraction property. The bias of exponential smoothing in policy evaluation can be characterized as:

$$\left\| v^\pi - v_\beta^\pi \right\|_\infty \leq \sum_{i=0}^\infty \gamma^i \left\| (P^i - ((1-\beta)P + \beta(1-\lambda) \sum_{j=1}^\infty \lambda^{j-1} \widetilde{P}^j)^i)r \right\|_\infty . \tag{11}$$

Using more powerful regularizers could be beneficial, for example to reduce variance by smoothing over more values (exponential smoothing) or to model the trend of the value function through the trajectory using trend adjusted model [13]. An example of a temporal policy evaluation with temporal regularization using exponential smoothing is provided in Algorithm 1.

---

**Algorithm 1** Policy evaluation with exponential smoothing

---

1: Input: $\pi, \alpha, \gamma, \beta, \lambda$
2: $p = v(s)$
3: **for all** steps **do**
4:    Choose $a \sim \pi(s)$
5:    Take action $a$, observe reward $r(s)$ and next state $s'$
6:    $v(s) = v(s) + \alpha(r(s) + \gamma((1 - \beta)v(s') + \beta p))$
7:    $p = (1 - \lambda)v(s) + \lambda p$
8: **end for**

---

**Control case:**   Temporal regularization can be extended to MDPs with actions by modifying the target of the value function (or the Q values) using temporal regularization. Experiments (Sec. 5.6) present an example of how temporal regularization can be applied within an actor-critic framework. The theoretical analysis of the control case is outside the scope of this paper.

**Temporal difference with function approximation:**   It is also possible to extend temporal regularization using function approximation such as semi-gradient TD [33]. Assuming a function $v_\theta^\beta$ parameterized by $\theta$, we can consider $r(s) + \gamma((1 - \beta)v_\theta^\beta(s_{t+1}) + \beta v_\theta^\beta(s_{t-1})) - v_\theta^\beta(s_t)$ as the target and differentiate with respect to $v_\theta^\beta(s_t)$. An example of a temporally regularized semi-gradient TD algorithm can be found in the appendix.

## 5   Experiment

We now presents empirical results illustrating potential advantages of temporal regularization, and characterizing its bias and variance effects on value estimation and control.

### 5.1   Mixing time

This first experiment showcases that the underlying Markov chain of a MDP can have a smaller mixing time when temporally regularized. The mixing time can be seen as the number of time steps required for the Markov chain to get *close enough* to its stationary distribution. Therefore, the mixing time also determines the rate at which policy evaluation will converge to the optimal value function [3]. We consider a synthetic MDP with 10 states where transition probabilities are sampled from the uniform distribution. Let $P^\infty$ denote a matrix where columns consists of the stationary distribution $\mu$. To compare the mixing time, we evaluate the error corresponding to the distance of $P^i$ and $\left((1 - \beta)P + \beta\widetilde{P}\right)^i$ to the convergence point $P^\infty$ after $i$ iterations. Figure 1 displays the error curve when varying the regularization parameter $\beta$. We observe a U-shaped error curve, that intermediate values of $\beta$ in this example yields faster mixing time. One explanation is that transition matrices with extreme probabilities (low or high) yield poorly conditioned transition matrices. Regularizing with the reversal Markov chain often leads to a better conditioned matrix at the cost of injecting bias.

### 5.2   Bias

It is well known that reducing variance comes at the expense of inducing (smaller) bias. This has been characterized previously (Sec. 4) in terms of the difference between the original Markov chain and the reversal weighted by the reward. In this experiment, we attempt to give an intuitive idea of what this means. More specifically, we would expect the bias to be small if values along the trajectories have similar values. To this end, we consider a synthetic MDP with 10 states where both transition functions and rewards are sampled randomly from a uniform distribution. In order to create temporal dependencies in the trajectory, we smooth the rewards of $N$ states that are temporally close (in terms of trajectory) using the following formula: $r(s_t) = \frac{r(s_t) + r(s_{t+1})}{2}$. Figure 2 shows the difference between the regularized and un-regularized MDPs as $N$ changes, for different values of regularization parameter $\beta$. We observe that increasing $N$, meaning more states get rewards close

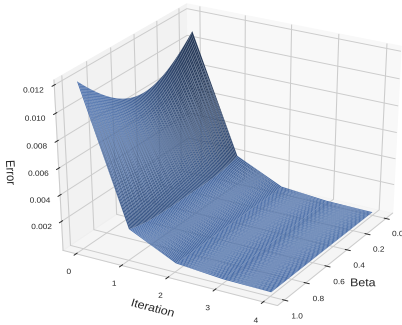

Figure 1: Distance between the stationary transition probabilities and the estimated transition probability for different values of regularization parameter $\beta$.

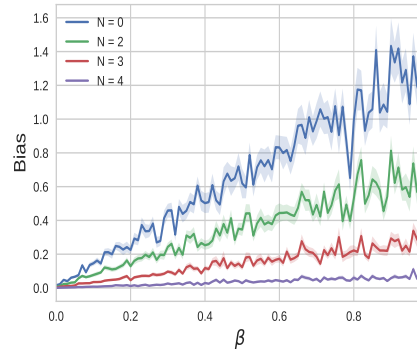

Figure 2: Mean difference between $v_\beta^\pi$ and $v^\pi$ given the regularization parameter $\beta$, for different amount of smoothed states $N$.

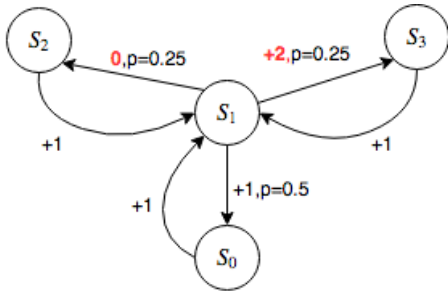

Figure 3: Synthetic MDP where state $S_1$ has high variance.

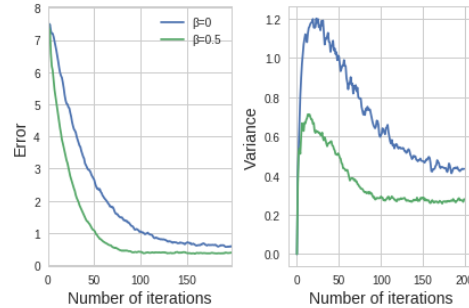

Figure 4: Left plot shows absolute difference between original ($v^\pi(S_1)$) and regularized ($v_\beta^\pi(S_1)$) state value estimates to the optimal value $v^*(S_1)$. Right plot shows the variance of the estimates $v$.

to one another, results into less bias. This is due to rewards putting emphasis on states where the original and reversal Markov chain are similar.

## 5.3 Variance

The primary motivation of this work is to reduce variance, therefore we now consider an experiment targeting this aspect. Figure 3 shows an example of a synthetic, 3-state MDP, where the variance of $S_1$ is (relatively) high. We consider an agent that is evolving in this world, changing states following the stochastic policy indicated. We are interested in the error when estimating the optimal state value of $S_1$, $v^*(S_1)$, with and without temporal regularization, denoted $v_\beta^\pi(S_1)$, $v^\pi(S_1)$, respectively.

Figure 4 shows these errors at each iteration, averaged over 100 runs. We observe that temporal regularization indeed reduces the variance and thus helps the learning process by making the value function easier to learn.

## 5.4 Propagation of the information

We now illustrate with a simple experiment how temporal regularization allows the information to spread faster among the different states of the MDP. For this purpose, we consider a simple MDP, where an agent walks randomly in two rooms (18 states) using four actions (up, down, left, right), and a discount factor $\gamma = 0.9$. The reward is $r_t = 1$ everywhere and passing the door between rooms (shown in red on Figure 5) only happens 50% of the time (on attempt). The episode starts at the

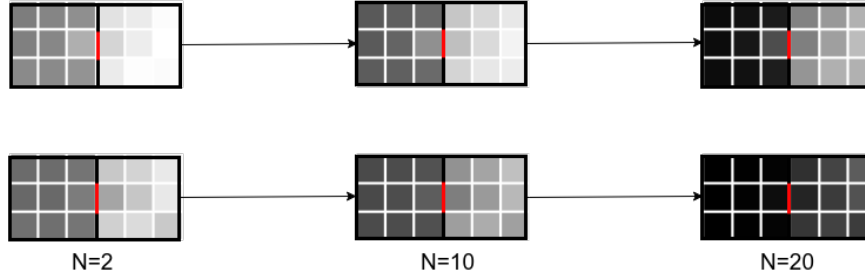

N=2                N=10                N=20

Figure 5: Proximity of the estimated state value to the optimal value after $N$ trajectories. Top row is the original room environment and bottom row is the regularized one ($\beta = 0.5$). Darker is better.

top left and terminates when the agent reaches the bottom right corner. The sole goal is to learn the optimal value function by walking along this MDP (this is not a race toward the end).

Figure 5 shows the proximity of the estimated state value to the optimal value with and without temporal regularization. The darker the state, the closer it is to its optimal value. The heatmap scale has been adjusted at each trajectory to observe the difference between both methods. We first notice that the overall propagation of the information in the regularized MDP is faster than in the original one. We also observe that, when first entering the second room, bootstrapping on values coming from the first room allows the agent to learn the optimal value faster. This suggest that temporal regularization could help agents explore faster by using their prior from the previous visited state for learning the corresponding optimal value faster. It is also possible to consider more complex and powerful regularizers. Let us study a different time series prediction model, namely exponential averaging, as defined in (10). The complexity of such models is usually articulated by hyper-parameters, allowing complex models to improve performance by better adapting to problems. We illustrate this by comparing the performance of regularization using the previous state and an exponential averaging of all previous states. Fig. 6 shows the average error on the value estimate using past state smoothing, exponential smoothing, and without smoothing. In this setting, exponential smoothing transfers information faster, thus enabling faster convergence to the true value.

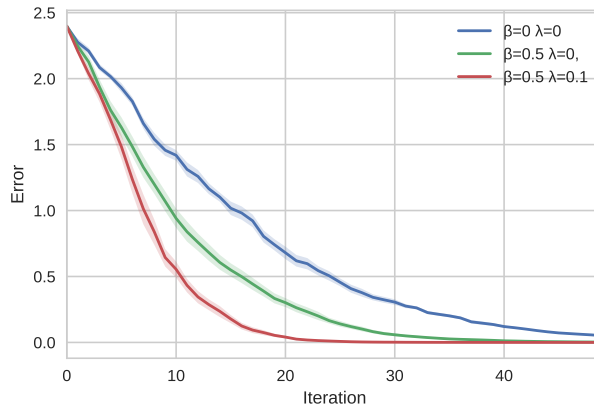

Figure 6: Benefits of complex regularizers on the room domain.

## 5.5 Noisy state representation

The next experiment illustrates a major strength of temporal regularization, that is its robustness to noise in the state representation. This situation can naturally arise when the state sensors are noisy or insufficient to avoid aliasing. For this task, we consider the synthetic, one dimensional, continuous setting. A learner evolving in this environment walks randomly along this line with a discount factor $\gamma = 0.95$. Let $x_t \in [0, 1]$ denote the position of the agent along the line at time $t$. The next position $x_{t+1} = x_t + a_t$, where action $a_t \sim \mathcal{N}(0, 0.05)$. The state of the agent corresponds to the position

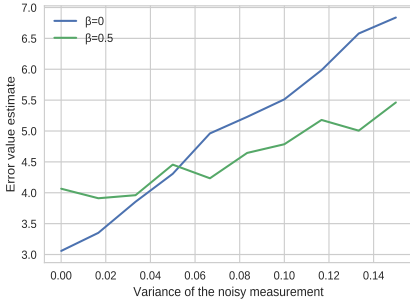
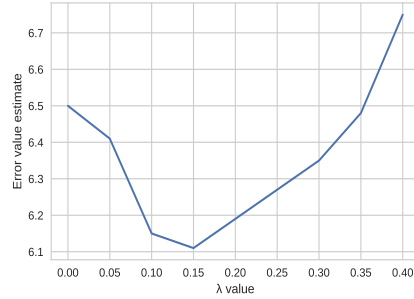

Figure 7: Absolute distance from the original ($\theta^\pi$) and the regularized ($\theta^\pi_\beta$) state value estimates to the optimal parameter $\theta^*$ given the noise variance $\sigma^2$ in state sensors.

Figure 8: Impact of complex regularizer parameterization ($\lambda$) on the noisy walk using exponential smoothing.

perturbed by a zero-centered Gaussian noise $\epsilon_t$, such that $s_t = x_t + \epsilon_t$, where $\epsilon_t \sim \mathcal{N}(0, \sigma^2)$ are i.i.d. When the agent moves to a new position $x_{t+1}$, it receives a reward $r_t = x_{t+1}$. The episode ends after 1000 steps. In this experiment we model the value function using a linear model with a single parameter $\theta$. We are interested in the error when estimating the optimal parameter function $\theta^*$ with and without temporal regularization, that is $\theta^\pi_\beta$ and $\theta^\pi$, respectively. In this case we use the TD version of temporal regularization presented at the end of Sec. 4. Figure 7 shows these errors, averaged over 1000 repetitions, for different values of noise variance $\sigma^2$. We observe that as the noise variance increases, the un-regularized estimate becomes less accurate, while temporal regularization is more robust. Using more complex regularizer can improve performance as shown in the previous section but this potential gain comes at the price of a potential loss in case of model misfit. Fig. 8 shows the absolute distance from the regularized state estimate (using exponential smoothing) to the optimal value while varying $\lambda$ (higher $\lambda$ = more smoothing). Increasing smoothing improves performance up to some point, but when $\lambda$ is not well fit the bias becomes too strong and performance declines. This is a classic bias-variance tradeoff. This experiment highlights a case where temporal regularization is effective even in the absence of smoothness in the state space (which other regularization methods would target). This is further highlighted in the next experiments.

## 5.6   Deep reinforcement learning

To showcase the potential of temporal regularization in high dimensional settings, we adapt an actor-critic based method (PPO [28]) using temporal regularization. More specifically, we incorporate temporal regularization as exponential smoothing in the target of the critic. PPO uses the general advantage estimator $\hat{A}_t = \delta_t + \gamma\lambda\delta_{t+1} + ... + (\gamma\lambda)^{T-t+1}\delta_T$ where $\delta_t = r_t + \gamma v(s_{t+1}) - v(s_t)$. We regularize $\delta_t$ such that $\delta_t^\beta = r_t + \gamma((1-\beta)v(s_{t+1}) + \beta\widetilde{v}(s_{t-1}))) - v(s_t)$ using exponential smoothing $\widetilde{v}(s_t) = (1-\lambda)v(s_t) + \lambda\widetilde{v}(s_{t-1})$ as described in Eq. (10). $\widetilde{v}$ is an exponentially decaying sum over all $t$ previous state values encountered in the trajectory. We evaluate the performance in the Arcade Learning Environment [4], where we consider the following performance measure:

$$\frac{\text{regularized} - \text{baseline}}{\text{baseline} - \text{random}}. \tag{12}$$

The hyper-parameters for the temporal regularization are $\beta = \lambda = 0.2$ and a decay of $1\mathrm{e}^{-5}$. Those are selected on 7 games and 3 training seeds. All other hyper-parameters correspond to the one used in the PPO paper. Our implementation[2] is based on the publicly available OpenAI codebase [8]. The previous four frames are considered as the state representation [22]. For each game, 10 independent runs (10 random seeds) are performed.

The results reported in Figure 9 show that adding temporal regularization improves the performance on multiple games. This suggests that the regularized optimal value function may be smoother and thus easier to learn, even when using function approximation with deep learning. Also, as shown in

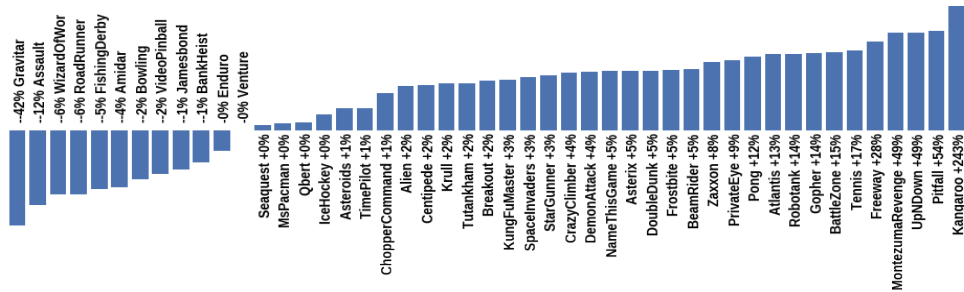

Figure 9: Performance (Eq. 12) of a temporally regularized PPO on a suite of Atari games.

previous experiments (Sec. 5.5), temporal regularization being independent of spatial representation makes it more robust to mis-specification of the state features, which is a challenge in some of these games (e.g. when assuming full state representation using some previous frames).

## 6 Discussion

**Noisy states:** Is is often assumed that the full state can be determined, while in practice, the Markov property rarely holds. This is the case, for example, when taking the four last frames to represent the state in Atari games [22]. A problem that arises when treating a partially observable MDP (POMDP) as a fully observable is that it may no longer be possible to assume that the value function is smooth over the state space [31]. For example, the observed features may be similar for two states that are intrinsically different, leading to highly different values for states that are nearby in the state space. Previous experiments on noisy state representation (Sec. 5.5) and on the Atari games (Sec. 5.6) show that temporal regularization provides robustness to those cases. This makes it an appealing technique in real-world environments, where it is harder to provide the agent with the full state.

**Choice of the regularization parameter:** The bias induced by the regularization parameter $\beta$ can be detrimental for the learning in the long run. A first attempt to mitigate this bias is just to decay the regularization as learning advances, as it is done in the deep learning experiment (Sec. 5.6). Among different avenues that could be explored, an interesting one could be to aim for a state dependent regularization. For example, in the tabular case, one could consider $\beta$ as a function of the number of visits to a particular state.

**Smoother objective:** Previous work [18] looked at how the smoothness of the objective function relates to the convergence speed of RL algorithms. An analogy can be drawn with convex optimization where the rate of convergence is dependent on the Lipschitz (smoothness) constant [6]. By smoothing the value temporally we argue that the optimal value function can be smoother. This would be beneficial in high-dimensional state space where the use of deep neural network is required. This could explain the performance displayed using temporal regularization on Atari games (Sec. 5.6). The notion of temporal regularization is also behind multi-step methods [32]; it may be worthwhile to further explore how these methods are related.

**Conclusion:** This paper tackles the problem of regularization in RL from a new angle, that is from a temporal perspective. In contrast with typical spatial regularization, where one assumes that rewards are close for nearby states in the state space, temporal regularization rather assumes that rewards are close for states *visited closely in time*. This approach allows information to propagate faster into states that are hard to reach, which could prove useful for exploration. The robustness of the proposed approach to noisy state representations and its interesting properties should motivate further work to explore novel ways of exploiting temporal information.

### Acknowledgments

The authors wish to thank Pierre-Luc Bacon, Harsh Satija and Joshua Romoff for helpful discussions. Financial support was provided by NSERC and Facebook. This research was enabled by support provided by Compute Canada. We thank the reviewers for insightful comments and suggestions.

## Footnotes

[1]$\mu_i$ defines the $i$th element of $\mu$

[2]The code can be found `https://github.com/pierthodo/temporal_regularization`.

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
