[Supplementary Material · supplementary.pdf]

# 7 Appendix

**Lemma 1.** *$P$ and $(1 - \beta)P + \beta\widetilde{P}$ have the same stationary distribution $\mu$   $\forall \beta \in [0, 1]$.*

*Proof.* It is known that $P^\pi$ and $\widetilde{P^\pi}$ have the same stationary distribution. Using this fact we have that

$$
\begin{aligned}
\mu((1 - \beta)P^\pi + \beta\widetilde{P^\pi}) &= (1 - \beta)\mu P^\pi + \beta\mu\widetilde{P^\pi} \\
&= (1 - \beta)\mu + \beta\mu \\
&= \mu.
\end{aligned}
\tag{13}
$$

$\square$

**Property 2.** *For a reward vector r, the MDP defined by the the transition matrix $P$ and $(1-\beta)P+\beta\widetilde{P}$ have the same discounted average reward $\rho$.*

$$
\frac{\rho}{1 - \alpha} = \sum_i^\infty \gamma^i \pi^T r.
\tag{14}
$$

*Proof.* Using lemma 1, both $P$ and $(1 - \beta)P + \beta\widetilde{P}$ have the same stationary distribution and so discounted average reward. $\square$

**Lemma 2.** *The convex combination of two row stochastic matrices is also row stochastic.*

*Proof.* Let e be vector a columns vectors of 1.

$$
\begin{aligned}
(\beta P^\pi + (1 - \beta)\widetilde{P^\pi})e &= \beta P^\pi e + (1 - \beta)\widetilde{P^\pi}e \\
&= \beta e + (1 - \beta)e \\
&= e.
\end{aligned}
\tag{15}
$$

$\square$

---

**Algorithm 2** Temporally regularized semi-gradient TD

---
1: Input: policy $\pi, \beta, \gamma$
2: **for all** steps **do**
3:    Choose $a \sim \pi(s_t)$
4:    Take action $a$, observe $r(s), s_{t+1}$
5:    $\theta = \theta + \alpha(r + \gamma((1 - \beta)v_\theta(s_{t+1}) + \beta v_\theta(s_{t-1})) - v_\theta(s_t))\nabla v_\theta(s_t)$
6: **end for**

---

Figure 10: Average reward per episode on Atari games.