[Reviews · NeurIPS 2018]

Reviewer 1



This paper is very interesting. One previous assumption in TD learning is that reward are close with states in proximity of the state space, which has been pointed out by many papers is not realistic and have problems for spatial value function regularization. Instead, this paper make the assumption that rewards are close for states. Overall this paper has a very good motivation, and the literature review shows that the author is knowledgable of this field. This paper could open a novel area of temporal regularization that received inadequate attention before. Here are my major concerns: 1. The author tries to squeeze too much content to show the effectiveness of the proposed method. For example, the experiments covers both value function approximation and control. To me, the effect of control by the improved value function approximation is indirect and less clear than showing results of policy evaluation. I would suggest the author to remove (or put it to the appendix) the control part (such as line 137-140, and section 5.6) and focus more on the value approximation part. 2. The comparison study of different temporal regularization methods is incomplete. As the author pointed out in line 128, different temporal regularization methods could be used. Then the comparison study of different methods are completely missing in the paper, which makes the paper incomplete from this perspective. 3 connections with previous work: This temporal regularization will serve a very similar purpose as Residual gradient (RG) vs. TD on backward bootstrapping, where the value is affected by both the previous state and the next state. The benefit is stability and the disadvantage of deteriorated prediction. This has been investigated in several previous papers, such as Sutton, R. S., Maei, H. R., Precup, D., Bhatnagar, S., Silver, D., Szepesvári, C., & Wiewiora, E. (2009, June). Fast gradient-descent methods for temporal-difference learning with linear function approximation. In Proceedings of the 26th Annual International Conference on Machine Learning (pp. 993-1000). ACM. (detailed discussed backward bootstrapping) Li, L. (2008, July). A worst-case comparison between temporal difference and residual gradient with linear function approximation. In Proceedings of the 25th international conference on machine learning (pp. 560-567). ACM. (compare the prediction and stability of TD and RG, though not explicitly bridging the connections with backward bootstrapping) Some minor suggestions: line 56-67: it is much better to give reasons and toy examples why reversal Markov chain property (Def 2) is less violated in practice than Detailed balance (Def. 1). line 110& 122. It is better to change ‘Property 1’ —> ‘Proposition 1’. line 124 Does this imply the average reward \rho is more robust than the value function V? what else does it imply here? line 135 reference for ’trend-ajusted model’ is missing. === I have read the response, and have updated my reviews.

Reviewer 2



The paper proposes a temporal regularization technique for the estimation of a value function. The idea consists in averaging over previous values weighted with the probabilities of a reversal Markov chain. Pros: - The idea of using the reversal Markov chain and related results (although simple) are novel, as far as I know - The experimental results are convincing - The paper is reasonably clear Cons: - The presentation could have been more rigorous. Notably, the notational issues (see below) should be fixed before publication - I would have appreciated more details on how PPO is extended with temporal regularization. Is (10) applied with a finite N? Minor remarks and typos: l.73: the notation \pi( . | s) doesn't fit its definition as a function from S x A (3): shouldn't r also depends on \pi? (4): r(a) is not defined, the max operator over vector should also be defined (5): a_t should also appear or the dependence on \pi should be clearer l.129-131 and (9): n vs N and several index problems l.135: missing word caption of Fig.4: missing space

Reviewer 3



The paper introduces a modification to the Bellman equation in which a relationship is enforced not only with the the next state in a trajectory, but also the previous state. This modified Bellman equation is shown to have the qualities we require of a Bellman equation, namely, it is a contraction, it converges to the value of the original Bellman equation. The paper then experimentally demonstrates its use on domains designed for its strengths, as well as more general Atari problems. On the more general problems, a temporally-regularized version of PPO results in an improvement over standard PPO on many examples. First, a complement: I am surprised this hasn't been done before. The writing is clear, and the motivation and results are explained well enough as to be obvious. My concern is that this is not likely to be that impactful. There are of course many modifications to the Bellman equation out there to address different small opportunities, which have been explained, nodded interestedly at by an audience, and then set aside. I suspect this will be the same. Many of those modifications were deservedly presented at ICML or NIPS; I'd like it if this was as well, but in today's highly competitive acceptance environment, there many be too many more-impactful submissions.